# High-Resolution Wastewater-Based Surveillance of Three Influenza Seasons (2022–2025) Reveals Distinct Seasonal Patterns of Viral Activity in Munich, Germany

**DOI:** 10.3390/microorganisms13112630

**Published:** 2025-11-20

**Authors:** Jessica Neusser, Astrid Zierer, Anna Riedl, Jasmin Javanmardi, Raquel Rubio-Acero, Elisabeth Waldeck, Thomas Kletke, Annemarie Bschorer, Stefanie Huber, Patrick Dudler, Martin Hoch, Merle M. Böhmer, Caroline Herr, Ute Eberle, Andreas Sing, Nikolaus Ackermann, Michael Hoelscher, Katharina Springer, Andreas Wieser

**Affiliations:** 1Institute of Infectious Diseases and Tropical Medicine, LMU University Hospital, Ludwig-Maximilians-Universität (LMU) München, 80802 Munich, Germany; jessica.neusser@lgl.bayern.de (J.N.); jasmin.javanmardi@med.uni-muenchen.de (J.J.); raquel.rubio@med.uni-muenchen.de (R.R.-A.); michael.hoelscher@med.uni-muenchen.de (M.H.); 2Bavarian Health and Food Safety Authority, 91058 Erlangen, Germany; astrid.zierer@lgl.bayern.de (A.Z.); anna.riedl@lgl.bayern.de (A.R.); annemarie.bschorer@lgl.bayern.de (A.B.); stefanie.huber@lgl.bayern.de (S.H.); patrick.dudler@lgl.bayern.de (P.D.); martin.hoch@lgl.bayern.de (M.H.); merle.boehmer@lgl.bayern.de (M.M.B.); caroline.herr@lgl.bayern.de (C.H.); ute.eberle@lgl.bayern.de (U.E.); andreas.sing@lgl.bayern.de (A.S.); nikolaus.ackermann@lgl.bayern.de (N.A.); katharina.springer@lgl.bayern.de (K.S.); 3Public Health Department Munich/Gesundheitsreferat Stadt München (GSR), 80335 Munich, Germany; elisabeth.waldeck@muenchen.de; 4Munich Metropolitan Sewer Authority/Münchner Stadtentwässerung (MSE), 81671 Munich, Germany; thomas.kletke@muenchen.de; 5Institute of Social Medicine and Health Systems Research, Otto-von-Guericke-University Magdeburg, 39120 Magdeburg, Germany; 6Institute and Outpatient Clinic for Occupational, Social and Environmental Medicine, LMU University Hospital, Ludwig-Maximilians-Universität (LMU) München, 80336 Munich, Germany; 7Pettenkofer School of Public Health, Chair of Public Health and Health Services Research, Ludwig-Maximilians-Universität (LMU) München, 81377 Munich, Germany; 8Comprehensive Pneumology Center Munich (CPC-M), Member of the German Center for Lung Research (DZL), 81377 Munich, Germany; 9German Center for Infection Research (DZIF), 80337 Munich, Germany; 10Fraunhofer Institute for Translational Medicine and Pharmacology ITMP, Immunology, Infection and Pandemic Research IIP, 80799 Munich, Germany; 11Unit Global Health, Helmholtz Zentrum München, German Research Center for Environmental Health (HMGU), 85764 Neuherberg, Germany; 12Max von Pettenkofer Institute, Faculty of Medicine, Ludwig-Maximilians-Universität (LMU) München, 80336 Munich, Germany

**Keywords:** influenza, influenza A virus (IAV), influenza B virus (IBV), wastewater, sewage, wastewater-based surveillance, wastewater-based epidemiology, grab sample, composite sample

## Abstract

In the Northern Hemisphere, annual waves of influenza disease with varying degrees of spread and severity are observed each winter. With wastewater-based surveillance (WBS), including both centralized (one wastewater treatment plant, WWTP) and decentralized (three sewers) sampling, we aimed to detect differences in influenza viral copy numbers in wastewater over time, to investigate (sub)-community transmission within a city. A total of 313 grab/spot and composite samples were collected in Munich, Germany, during three consecutive influenza seasons (2022/23, 2023/24, and 2024/25) and were analyzed for influenza A virus (IAV) and influenza B virus (IBV) nucleic acids using digital droplet PCR (ddPCR). IAV and IBV wastewater copy numbers and citywide reported influenza cases showed strong correlations in both sampling approaches, suggesting the decentralized approach to be a reliable indicator of infection trends across the city. The three influenza seasons analyzed differed significantly in terms of their seasonal distribution, for example, exhibiting a strong co-circulation of IAV and IBV only in the 2024/25 season. Only with wastewater analysis, we reveal a reporting delay of influenza A cases at the beginning of the 2023/24 season. Higher influenza copy numbers were detected in sewer samples compared to the WWTP influent, likely due to viral decay. The study underscores the potential of influenza WBS to enable detection of seasonal onset early, identify local transmission patterns, and reveal underreporting in routine surveillance systems.

## 1. Introduction

In recent years, wastewater-based surveillance (WBS) has primarily concentrated on severe acute respiratory syndrome coronavirus type 2 (SARS-CoV-2) surveillance, achieving substantial public visibility. Nevertheless, a more comprehensive epidemiological understanding of other respiratory infectious viruses is also warranted given the public health interest. Influenza A and B viruses (IAV and IBV) have historically posed the most significant seasonal threat in terms of morbidity and mortality, with an estimated number of about 50 million influenza cases per season and up to 70,000 deaths per year in Europe alone. Influenza viruses cause annual epidemic outbreaks with severe respiratory symptoms primarily in high-risk populations like small children, pregnant women, the elderly, the immunosuppressed, and patients with comorbidities [1]. After the coronavirus disease 2019 (COVID-19) pandemic, IAV and IBV showed altered infection patterns, such as an early and sharp increase in case numbers, possibly due to the termination of pandemic-related infection control measures in Germany [2]. The unanticipated surge of infections in 2022, which led to the overwhelming of numerous medical facilities across the Northern Hemisphere, underscores the necessity for more comprehensive surveillance systems in the future [3,4].

Surveillance of IAV and IBV is traditionally based on multiple data sources, including sentinel surveillance data from selected primary care facilities, hospitalization records, and notification data. A major challenge is that the clinical data accessible for analysis is often restricted and suffers from bias introduced by factors such as testing capacity and the prevalence of associated clinical manifestations of the circulating strain. Notification data is subject to inherent limitations and temporal delays arising from the complexity of the reporting infrastructure. Generally, only the start and the end of the season are estimated, but the disease burden of influenza is underestimated [5]. The Robert Koch Institute (RKI) in Germany posits that only a small proportion of influenza cases, around 1%, result in physician consultations [6]. WBS as a component of an integrated surveillance strategy, has the potential to augment established monitoring systems, a capability demonstrated previously for SARS-CoV-2 [7].

Although fecal shedding may not be observed in every influenza case [8], a single wastewater sample covers a significant percentage of the population, as it contains a pooled sample of multiple excretions (stool, urine, saliva, mucus, and sputum), thereby enhancing the probability of identifying cases. WBS studies on influenza viruses demonstrate, that viral RNA exhibits acceptable stability in both solid wastewater components [9,10,11], and the liquid phase of wastewater [12,13,14,15,16], which was utilized in the present study.

Many influenza WBS studies focused on a centralized wastewater sampling strategy [17,18,19] using WWTP samples. The role of a decentralized sampling strategy (on a building or sewer level) for influenza virus surveillance should also be examined more closely, as data availability is so far limited [13,20,21,22]. This method, which has also been employed during the COVID-19 pandemic for SARS-CoV-2 [23,24,25,26], facilitates a rapid public health response to atypical outbreaks of viruses. For instance, it can be utilized in the event of an outbreak of a novel pathogen with pandemic potential. In decentralized wastewater sampling strategies, the use of composite samples and qualified spot/grab samples needs to be studied for influenza, as the role of viral decay for influenza has not yet been investigated. Decentralized sampling is considerably more complex, characterized by limited accessibility to sewage due to the absence of electricity and readily accessible manholes in critical areas. So far, mainly WWTP samples have been analyzed for differences in grab/composite sampling [27,28,29,30].

In this manuscript, we present data from 313 samples collected in three consecutive influenza seasons (2022/23, 2023/24, and 2024/25) as part of a sewer specific and municipal wastewater monitoring study in Munich for influenza A and B viruses. The dataset includes viral copy numbers combined with sentinel surveillance data and citywide or sewer specific notification data from laboratory-confirmed influenza A and B cases. By using high-resolution surveillance data, the different dynamics of infection events within cities can be tracked in more detail. This methodology may prove to be of particular value in larger cities and settings where the availability of centralized WWTP samples is limited.

## 2. Materials and Methods

### 2.1. Wastewater Sampling

Selected sampling points in Munich have been used in COVID-19 wastewater monitoring for several years. Transport, processing, and storage of the wastewater samples were carried out according to our established protocol, which has been described elsewhere [7]. Different types of wastewater samples were analyzed as part of the study during influenza season from November to April (season 2022/23–2024/25). Qualified spot samples of 500 mL were taken twice a week, on Tuesdays and Fridays between 10:00 a.m. and 11:00 a.m. during the morning flush from the sewer catchment areas of three city districts (sampling sites named Schmidbartlanger, Gysslinger Becken, and Schenkendorfstr.) during season 2022/23.

To mitigate potential viral decay and sedimentation compounded by protracted travel times, sampling sites for spot sampling were carefully selected, ensuring that the maximum flow time from sink to sample did not exceed five hours. The selected spot sampling sites adhered to strict criteria: Catchment areas were required to exclude industrial complexes and large hospitals, maintain common foreign water influx below 20%, and serve a population significantly exceeding 10,000 inhabitants to circumvent potential privacy and data protection issues. We also excluded sampling sites with significant collateral sewage draining pipes to be able to define the exact catchment area. Moreover, the regions were chosen as primarily residential areas with an easily accessible location for spot sampling. Three subsamples of the same volume were collected every 10 min and pooled for a qualified spot sample. For the remaining seasons, only weekly sampling was possible in the Schmidbartlanger sewer. For the comparison of composite and qualified samples, 12 weekly paired measurements were carried out in the 2024/25 season in the Schmidbartlanger sewer, and in the 2022/23 season, 32 weekly paired measurements were carried out in the WWTP.

In addition, two 1 L 24 h composite samples per week (Mondays/Tuesdays and Thursdays/Fridays) and corresponding qualified spot samples were taken from the untreated influent of the main wastewater treatment plant (WWTP) Gut Grosslappen in Munich. An automated sampler took a sample from 8:00 a.m. onwards in 10 min time intervals to capture the morning flush surge, combining a volume- and time proportional sampling method. In Munich, two large WWTPs are operated; around 60% of the wastewater treated in Munich is received by Gut Grosslappen WWTP as the city’s largest WWTP. This WWTP serves about 961,000 inhabitants of the metropolitan area, which corresponds to approximately 64% of Munich’s population. The specific sampling areas and characteristics are shown in Figure 1 and Table 1.

### 2.2. Virus Concentration and Nucleic Acid Extraction

The virus purification in fresh wastewater samples followed our established protocol for SARS-CoV-2 surveillance [7]. The 45 mL samples of fresh wastewater were initially centrifuged at 3000× *g* using centrifuge tubes (Cat. No. 430829, Corning Science Mexico S.A. de C.V., Reynosa, Mexico) at 4 °C for 20 min to eliminate debris and bacteria. From each sample, 38 mL of the supernatant was transferred to ultracentrifuge tubes (Cat. No. 3138–0050, Thermo Scientific Nalgene, Schwerte, Germany) and subjected to further centrifugation at 26,000× *g* at 4 °C for 1 h. Nucleic acids from the resulting pellet were extracted with a magnetic-based isolation method using the nucleic acid extractor TANBead Maelstrom 4810 (Taiwan Advanced Nanotech Inc., Guishan, Taiwan). The pellets designated for automated extraction were reconstituted in 300 µL of nuclease-free water (Cat. No. 436912C, VWR Chemicals, Darmstadt, Germany) and mixed with 10 µL of proteinase K provided with the kit “TANBead Optipure Viral Auto Tube” (Cat. No. M665S46, TANBead Nucleic Acid Extraction Kit; Taiwan Advanced Nanotech Inc., Guishan, Taiwan) prior to extraction. The extraction process was carried out using the “667 rapid”-program pre-programmed in the Maelstrom 4810 instrument. Total nucleic acids were eluted in 70 µL of the elution buffer, and the eluates were stored at −80 °C prior to influenza analysis by ddPCR or real-time quantitative PCR (RT-qPCR) for PMMoV measurements.

### 2.3. Quantification of Influenza Viruses with Digital Droplet PCR (ddPCR)

A low limit of detection and the most accurate quantification of pathogen targets are essential for WBS. For this reason, in this study ddPCR was performed, offering benefits compared to RT-qPCR due to its increased analytical sensitivity, high reproducibility of results, and, in particular, robustness against PCR inhibitors. Absolute viral quantification was performed using CDC recommended primers targeting the matrix (M)-gene for IAV and the nonstructural (NS)-gene for IBV [31]. The limit of detection (LoD) was determined by performing single ddPCR assays for each target using the primers and probes described in Appendix A Table A1. The LoD is defined as the lowest dilution of gene copies in a PCR reaction that is theoretically detectable using optimized methods. The template employed in this study was purified RNA of influenza A and B viruses (Cat. No. MBC028-30, Amplirun^®^ Influenza RNA Control, Vircell Microbiologists, Granada, Spain). For this purpose, the dilution levels were selected so that they were around the expected detection limit. The gene copy numbers investigated were 1, 2.5, 5, 10, and 20 per reaction. Each dilution was measured in ten replicates, and the LOD corresponds to the lowest copy number at which all replicates were still positive. The LoD for IAV is 2.5 gene copies/reaction and for IBV 3.0 gene copies/reaction.

In summary, 5 μL of template RNA were added to 5 μL of One-Step RT-ddPCR advanced kit for probes (Bio-Rad Laboratories, Munich, Germany), 2 μL of reverse transcriptase (Bio-Rad; final concentration, 20 U/μL), 1 μL of dithiothreitol (DTT) (Bio-Rad; final concentration, 15 nM), 1,5 μL of primer and probe mix (final concentrations: primers, 900 nM; probe, 250 nM), and 4 μL of nuclease-free water (Qiagen, Hilden, Germany). Droplets were generated using a QX600 ddPCR droplet generator (Bio-Rad Laboratories, Munich, Germany). PCR was performed on a Mastercycler X50S (Eppendorf, Hamburg, Germany) with the following thermal conditions: reverse transcription (RT) at 50 °C for 60 min, enzyme activation at 95 °C for 10 min, then 40 cycles of a two-step program of denaturation at 95 °C for 30 s and annealing/extension at 58 °C for 1 min. Final enzyme inactivation was performed at 98 °C for 10 min, and thereafter, the samples were cooled down to 4 °C. After incubation at 4 °C for at least 30 min, droplets were analyzed using a QX600 Droplet Digital PCR System (Bio-Rad Laboratories, Munich, Germany), and the QX manager software (Bio-Rad Laboratories, Munich, Germany). RNA eluates showing less than 15,000 droplets were excluded from the analysis.

### 2.4. Quantification of Human Fecal Indicator PMMoV

Pepper Mild Mottle Virus (PMMoV), an abundant plant virus excreted in human feces, is widely utilized as a normalization standard for quantifying fecal load in wastewater surveillance. The quantification of PMMoV in wastewater samples was conducted using the RT-qPCR method, following the manufacturer’s protocol outlined in the GoTaq^®^ Enviro PMMoV Quant Kit, Quasar 670 (Cat. No. AM2140, Promega, Madison, WI, USA). In brief, 5 µL of a 1:10 sample dilution was combined with 10 µL of Master Mix, 1 µL of primer/probe mix, 3.6 µL of nuclease-free water, and 0.4 µL of enzyme mix. The standard thermal cycle program with fluorophore Quasar 670/Cy5 detection was employed across 40 cycles, using the CFX96 Touch Real-Time PCR Detection System (Bio-Rad Laboratories, Hercules, CA, USA).

### 2.5. Data Sources and Calculation

#### 2.5.1. Notification Data

Influenza notification data according to the local infection protection law (IfSG § 7) was obtained from the City of Munich. Section 7 of the law specifies the notification requirements for laboratory-confirmed cases of notifiable diseases. This data comprises all laboratory-confirmed, differentiable influenza cases, aggregated on a weekly basis. The Munich Health Department (GSR) also provided notification data, broken down by sewer catchment area using shapefiles in ArcGIS using version 3.3.4. With this, we ensure that the reported case numbers also match the persons recorded in the wastewater. R version 4.3.1 (using R Studio) was used with packages fedmatch and arcgisbinding. To analyze the number of cases in wastewater catchment areas, the reported cases were first intersected with the address coordinates of these areas using a GIS-based procedure. After successful matching, the reported cases were merged with the wastewater catchment areas in order to calculate the number of cases per day and catchment area. It was not possible to differentiate between influenza A and B cases for the sewer specific notification data until calendar week 50 in 2023, as the information was not recorded prior. Due to the low number of cases in the sewer catchment areas, no distinction was made between influenza A and B cases.

#### 2.5.2. Primary Care Sentinel Surveillance Data

The Bavarian Influenza + Corona Sentinel (BIS + C) of the Bavarian Health and Food Safety Authority (LGL) examines nasal and throat swabs selected randomly from patients with acute respiratory diseases (ARE) who have been treated, in general, in practitioners’ (GP) or pediatricians’ practices in Bavaria. The number of medical practices per district or independent city within the seven Bavarian administrative districts corresponds as closely as possible to the local population density and age distribution in order to achieve a statistical approximation of the infection incidence of ARE-associated viral pathogens in the German federal state of Bavaria. An average of 31 medical practices in Munich took part, submitting four samples each week (optional two additional samples from patients over the age of 60) per participating GP practice. Data points with a total number of less than 20 tests per week were removed. The virological laboratory diagnostics of all BIS + C samples includes testing for influenza A and B viruses, including subtypes or lineages, for SARS-CoV-2 viruses and associated virus variants via variant-specific PCR (vPCR) followed by NGS, as well as testing for RSV A and B viruses. The positivity rate was also calculated as the proportion of throat swabs testing positive for any influenza virus strain among all samples examined by LGL. Aggregated sentinel surveillance data on influenza A and B from the city of Munich were compared with citywide notification data.

#### 2.5.3. Normalization of Wastewater Data

Target pathogen concentrations in wastewater are subject to dilution by rain, melting snow, foreign water influx, and fluctuating leaks. A substantial share of the existing sewer infrastructure in Munich consists of historically constructed combined sewers conveying both stormwater and domestic wastewater. Each sampling site presents unique hydraulic and wastewater characteristics, leading to inter-site variability. Flow rate measurements could not be performed due to technical constraints within the sewer system. Therefore, virus concentrations should be normalized to a stable, highly abundant fecal indicator such as PMMoV to account for the observed dilutions. A raw normalization factor was calculated for each sample by dividing the concentration of the target gene by the concentration of the fecal indicator, which represents the unitless influenza viral load per unit of fecal matter. For enhanced temporal comparability at each site, the PMMoV corrected ratio was rescaled to the concentration unit (copies/L). This process involved multiplying the ratio by the long-term, site specific median PMMoV concentration on 30–50 days with less than 3 mm/m^2^ precipitation, representing days of no or negligible precipitation. We applied the formula in the GoTaq^®^ Enviro PMMoV Quant Kit, Quasar 670 (Cat. No. AM2140, Promega, Madison, WI, USA) to quantitate the amount of PMMoV nucleic acid in a sample and also for quantitating influenza A and B nucleic acid.

#### 2.5.4. Statistical Analysis

All statistical analyses were computed using Prism GraphPad version 10 or R version 4.5.1 (using R Studio) and packages tidyverse, ggplot2, and fANCOVA. LOESS smoothing curves (and confidence intervals) were calculated based on log10-transformed data following RKI protocols [32]. LOESS smoothing is based on a local polynomial of degree two, selecting a smoothing parameter automatically according to the criterion of generalized cross validation individually in each constellation. To account for variations within values below the detection limit, a uniform distribution between zero and the LoD was assumed. We used the log-transformed wastewater copy number determined by LOESS smoothing on a daily basis to calculate Pearson correlation coefficients *r* between different sampling sites. Notification data are only available on an aggregated weekly basis. To assess correlations between wastewater copy numbers and weekly influenza notification data for Munich, we used the log-transformed wastewater copy numbers determined by LOESS on Wednesdays of the corresponding calendar week to represent average infection dynamics of the week. Regarding all Pearson correlation coefficients, confidence intervals and test decisions (two-tailed testing) were based on a significance level of α = 0.05. Graphs were created with R Studio. Pearson correlation coefficients and confidence intervals between different clinical surveillance data sources were computed with Prism GraphPad. Violin plots with embedded box plots were generated to compare the distribution of the (log)-ratios of spot and composite samples from Schmidbartlanger sewer and WWTP for influenza A and B, respectively. The Wilcoxon matched pairs signed rank test was conducted using Prism GraphPad to compare differences in viral copy numbers of the composite and qualified spot samples of sewer Schmidbartlanger and WWTP influent. As null hypothesis we stated that there is no difference in viral copy numbers. A two-tailed test at a significance level of α = 0.05 based on the exact distribution was performed. The graphs for comparison of spot and composite samples, primary care case data, citywide and sewer specific notification data were created using Prism GraphPad.

## 3. Results and Discussion

### 3.1. Differences in Influenza Wastewater Copy Numbers and Development over Time with Respect to Decentralized and Centralized Wastewater Sampling Strategy

For this study, wastewater samples from decentralized sampling sites, including three sewers, were examined. A total of 71 qualified spot samples and 12 composite samples from Schmidbartlanger, 37 samples from Gysslinger Becken, 36 samples from Schenkendorfstr., 125 composite and 32 qualified spot samples from the centralized sampling site WWTP Gut Grosslappen, Munich, were analyzed. Throughout the 2022/23 season, we analyzed IAV and IBV copy numbers in wastewater from multiple sewer sites as well as the WWTP (see Figure 2).

We observed different median levels of influenza copy numbers in wastewater depending on the sampling locations, which differed by a factor of 6 (equivalent to a difference of 0.78 at log scale; min. median IAV = 374 copies/L WWTP, max. median IAV = 2418 copies/L Schmidbartlanger; min. median IBV = 551 copies/L WWTP, max. median IBV = 3381 copies/L Schmidbartlanger). In particular, samples from the WWTP showed relatively low influenza copy numbers. In contrast, higher copy numbers were measured in the sewers, as shown in Appendix A, Table A2.

Analyzing development over time as shown in Figure 2, the influenza A wastewater copy numbers of two sampling sites (WWTP and Schmidbartlanger) show a simultaneous peak in viral concentrations at the beginning of December, while two others (Gysslinger Becken, Schenkendorfstr.) show a slightly delayed peak. A different pattern is observed for IBV copy numbers; the majority of sampling sites show a peak at the end of March/beginning of April. A Pearson’s correlation analysis including 95% confidence intervals was carried out for all pairwise measurements of influenza copy numbers between each sampling site during season 2022/23, as shown in Appendix A Table A3, with the result that all sampling sites are significantly positively correlated with each other. Correlations observed lie between *r* = 0.67 in sampling sites Schmidbartlanger-WWTP as well as Schenkendorfstr.-Schmidbartlanger and *r* = 0.86 in sampling sites Schenkendorfstr.-WWTP for IAV copy numbers. Correlations between *r* = 0.60 in Gysslinger Becken-WWTP and *r* = 0.98 in sewers Schenkendorfstr.-Schmidbartlanger were calculated for IBV copy numbers.

Examining influenza copy number levels over three consecutive influenza seasons at two selected sampling locations, WWTP and sewer Schmidbartlanger, as shown in Figure 3, it is noticeable that copy numbers of qualified spot samples of sewer Schmidbartlanger are higher (median IAV = 4338 copies/L; median IBV = 2499 copies/L) than the corresponding sample of WWTP (median IAV = 1220 copies/L; median IBV = 642 copies/L). The Wilcoxon matched pairs signed rank test indicated a significant difference in the amount of viruses present in the sewer Schmidbartlanger sample and in the WWTP sample (IAV and IBV: *p* < 0.0001, N = 66).

In the WWTP influent, there were more samples below the LoD for IAV 41% (N = 27/66) and IBV 53% (N = 35/66), in particular at the beginning and end of the season, compared to the samples from the sewer Schmidbartlanger for IAV 23% (N = 15/66) and IBV 33% (N = 22/66).

Similarly to other studies [12,14,16], our data also indicate a lower sensitivity of influenza RNA detection in WWTP influent samples compared to decentralized sewer samples (Figure 2). This could possibly be a consequence of prolonged flow times of up to 12 h in Munich and subsequent viral decay and sedimentation of virus in the WWTP samples. It has been shown that enveloped viruses in particular are inactivated more quickly in untreated municipal wastewater than non-enveloped viruses [33], suggesting consequences for influenza WES in samples with extended flow times. Strong dilution effects, which can occur in stormwater overflow basins (e.g., Gysslinger Becken or Schenkendorfstr. basin), different materials of piping, foreign water inflow, groundwater infiltration, or industry wastewater influx may also contribute to the variability of wastewater data between sewer districts [34].

Influenza monitoring may be more suitable at sampling points closer to wastewater production, such as samples with a maximum of 5 h of flow time from sink to sampling. 5 h were chosen to optimize the balance between maximizing the sampling area and minimizing viral decay to preserve sample integrity. The number of samples below the detection limit was also higher in the samples from the WWTP inflow compared to samples from the sewer Schmidbartlanger. Viral degradation is a plausible explanation for this observation. Higher viral loads in the sewer system could be interesting for potential influenza sequencing approaches. If a large part of the viral genome is fragmented and/or low concentrations in wastewater are found, as in the case of our wastewater treatment plant samples, analysis becomes more difficult. Decentralized sampling could provide a good alternative for this purpose. Distinct developments of influenza copy numbers in wastewater over time could also be a result of different infection numbers in the respective sampled regions in Munich, demonstrating the potential of a decentralized wastewater sampling approach. Overall, the large waves of infection can be mapped well by influenza copy numbers over time in all sampling locations, as shown in Figure 2, although individual sites show local and temporal variations in smaller outbreaks.

In general, IBV seems to be less frequently detected in wastewater studies than IAV [13,15,16,35,36], possibly due to rarer occurrence and faster decay. Our study and another German study found the opposite [12]. Notification data for this disease is also less reliable, which is presumably due to a lower rate of GP consultations due to milder disease manifestations. However, there are studies that suggest a similar severity of disease burden for influenza A and B [37]. This would also make IBV relevant for wastewater monitoring.

### 3.2. Differences in Copy Numbers of Qualified Spot Sample and Composite Sample Pairs in Decentralized (Schmidbartlanger Sewer) and Centralized Sampling Site Wastewater Treatment Plant

We were able to carry out weekly measurements taken on the same day, qualified spot samples, as well as 24 h composite samples, from the same location of sewer Schmidbartlanger for IAV and IBV (*n* = 12) during influenza season 2024/25. During the 2022/23 season, corresponding paired measurements could be obtained in the WWTP (*n* = 32). Figure 4a shows the differences in viral copy numbers in both sample types. In the decentralized wastewater sampling approach represented by the Schmidbartlanger sewer, there was no clear tendency towards a higher viral copy number in one of the sampling types. There was no hint for differences according to violin plots (Figure 4c) and the Wilcoxon matched pairs signed rank test (IAV: *p* = 0.85; IBV: *p* = 0.46, *n* = 12) in Schmidbartlanger sewer, although it should be noticed that sample size is small. According to the violin plot (Figure 4c) for WWTP IAV, the density distribution of the log-ratio is slightly shifted upward relative to zero, indicating a tendency toward higher copy numbers obtained by spot sampling, while showing higher variability. For WWTP IBV (see Figure 4c), most log-ratios are centered around zero, suggesting similar viral copy numbers obtained by both methods; however, a noticeable proportion of observations deviate substantially, reflecting discrepancies between the two sampling methods. The Wilcoxon matched pairs signed rank test revealed a significant difference for IAV, with higher viral loads in the spot sample for IAV (*p* = 0.008, *n* = 32), whereas no significant difference was observed for IBV (*p* = 0.06, *n*= 32).

Whereas Länsivaraa et al. [30] reported higher viral loads of non-enveloped viruses in composite samples than in grab samples from WWTP influents, our findings did not reveal a similar trend. Considering the substantial variation in viral loads across both sample types and locations and a small sample size, neither spot nor composite samples exhibited systematically higher viral loads. Augusto et al. found no differences between SARS-CoV-2 viral loads of composite and grab samples over 17 weeks in a WWTP [27]. Fragile, enveloped viruses may be prone to decay in composite samples. Qualified spot samples may not adequately reflect viral copy numbers due to their brief few-minute sampling period, in contrast to 24 h composite samples; however, this could not be confirmed in the present study. The time of spot sampling should be well chosen and cover the morning flush between 8 a.m. and 10 a.m., as most people use the toilet at this time, helping to collect a higher viral load [27].

### 3.3. High Concordance Across Three Consecutive Influenza Seasons (2022/23–2024/25) with Notification Data

To approach a more accurate estimate of infection incidence, two data sources for traditional influenza surveillance were compared over the years 2022–2025 (Figure 5). Firstly, laboratory-confirmed clinical influenza cases as part of the statutory reporting obligation per calendar week, and secondly, weekly data from the so-called outpatient sentinel surveillance, whose data basis comprises laboratory-confirmed influenza cases from selected outpatients’ practices in Munich, were obtained. The number of reported cases differed, but not the time resolved trends. The start and end of the influenza A and B seasons coincided, while more deviations between the two data sources were seen in the influenza B season, as the Pearson correlation coefficient suggests (influenza A: *r* = 0.90; influenza B: *r* = 0.61). Correlation coefficients with confidence intervals can be found in Appendix A, Table A4. We also compared sewer specific reporting data in two catchment areas with citywide influenza A and B cases, as shown in Figure 5c,d, in order to work out differences in the incidence of infection, with the result of a strong significant positive correlation (Pearson correlation Gysslinger Becken *r* = 0.99; Schmidbartlanger *r* = 0.96). The infection patterns observed in notification data were largely consistent between the sewer catchment areas and the city as a whole, with no major differences evident. As the citywide notification data included more cases, and was thus considered more stable, this data source was correlated with wastewater concentrations in the further analyses.

In Figure 3 we visualize the data from the comparison between a decentralized (Schmidbartlanger sewer, see Figure 3a) and centralized sampling strategy (WWTP, see Figure 3c). Viral copy numbers of three consecutive influenza seasons were analyzed (2022/23, 2023/24, and 2024/25) for both sampling sites in wastewater, and correlated with citywide reported influenza cases. Overall, smoothed influenza virus wastewater copy numbers of Schmidbartlanger correlated strongly with citywide notification data applying Pearson correlation coefficient (Figure 3a, IAV: *r* 22/23 = 0.87; *r* 23/24 = 0.64; *r* 24/25 = 0.71). Correlation coefficients with corresponding 95% confidence intervals are shown in Appendix A, Table A5. A significant positive correlation was observed for IBV wastewater copy numbers in Schmidbartlanger and citywide notification data (IBV: *r* 22/23 = 0.86; *r* 23/24 = 0.83; *r* 24/25 = 0.75).

The citywide influenza A case numbers correspond well with the LOESS smoothed viral copy numbers in the samples from the WWTP (Figure 3c, IAV: *r* 22/23 = 0.90; *r* 23/24 = 0.64; *r* 24/25 = 0.93). The same applies to the influenza B data (Figure 3c, IBV: *r* 22/23= 0.75; *r* 23/24 = 0.83; *r* 24/25= 0.89). In the second season, 2023/24, the correlation for IAV and clinical cases was weaker in both sampling sites, possibly due to the higher wastewater concentrations at the beginning of the season, while few cases were reported, as discussed in Section 3.4. In general, we observed in both sampling areas distinct seasonal patterns of IAV and IBV activity in the wastewater and notification data alike. Infection incidence in the Schmidbartlanger sewer site reflects the citywide infection incidence over three seasons. High Pearson correlation coefficients of influenza copy numbers between Schmidbartlanger and WWTP were observed across all seasons (Appendix A Table A3), with deviations observed only for IAV in the 2022/23 season (*r* = 0.67) and IBV in 2023/24 (*r* = 0.59). This indicates a similar infection pattern in both catchment areas.

### 3.4. Pronounced Inter-Seasonal Differences

The wave-like progressions of three consecutive (2022/23–2024/25) influenza A and B seasons could also be mapped in decentralized and centralized wastewater sampling sites, but varied depending on the season (Figure 3).

Season 1–2022/23

As shown in Figure 3a, in the first measured influenza season of 2022/23, we see a peak of IAV cases (calendar week 51/2022, *n* = 1201) and IAV wastewater copy numbers (13,939 copies/L WWTP; 93,422 and 63,990 copies/L Schmidbartlanger) at the beginning or in mid-December, whereas the seasons of 2023/24 and 2024/25 showed the highest number of IAV cases and peaks in IAV copy numbers at the end of January or in the first week of February. Due to limited social contacts and protective measures as a result of the COVID-19 pandemic, only low numbers of infections were recorded in the influenza seasons of 2020/21 [38] and 2021/22 in Germany [39]. In contrast, strong influenza activity occurred in 2022/23, characterized by an earlier start and peak of the season, as indicated in the notification and wastewater data of Figure 3. Influenza season 2022/23 was also the first post-pandemic period with a relaxation of restrictive measures [2,40].

Season 2—2023/24

We see in the data of season 2023/24 in Figure 3a,c for both sampling strategies and the citywide notification data an early increase in IAV copy numbers in wastewater even before case numbers indicated this (WWTP 11,185 copies/L in the first week of December with *n* = 28 reported influenza A cases). This was also reflected in the sewer specific wastewater data of Schmidbartlanger (34,373 copies/L on 5 December in Figure 3a). Clinical testing is particularly intensive at the beginning of the influenza season (starting from November), but the flu epidemic as defined by the RKI had not begun before the end of December [41]. The start of the influenza season can be detected early in wastewater, even earlier than in reporting data, as suggested by our observations at the beginning of the 2023/24 season. Groups of people who had not consulted a doctor, for example, very young patients, may have shed influenza. Wastewater data refers to the population level on an area basis and not to the individual level, bypassing biases such as health seeking behavior or oligosymptomatic cases potentially leading to massive underreporting.

Season 3—2024/25

The joint comparison of the influenza subtypes A and B shows that infections of both influenza virus strains occurred in parallel in season 2024/25, which was not the case in previous years (Figure 3a,c). A parallel circulation of influenza A and B can be influenced by various factors, including the natural variability of viruses and human behavior. After the SARS-CoV-2 pandemic, we first observed a spread of influenza A viruses; the infections of 2022/23 were caused by influenza A(H3N2), with only a small wave of influenza B/Victoria [40] [RKI, 2023 a]. A year later, in 2023/24, influenza A(H1N1)pdm09 was more likely to affect very young children (0–4) [42], but the wave probably did not fully unfold. In the 2024/2025 season, many schoolchildren (5–14 years) were particularly affected by influenza A(H1N1)pdm09, but to a large extent also by influenza B/Victoria [43]. It is therefore conceivable that there was greater immunity to IAV within the population and the IBV met with less resistance; seroprevalence data against B/Victoria before the start of the season also suggest this [44]. Overall, the match of the vaccines with circulating influenza strains was good, slightly higher for IAV than for IBV. Data on vaccine efficacy was not available. Moreover, influenza vaccination coverage in Germany has traditionally remained comparatively low; for instance, during the 2023/24 season, 38.2% of individuals aged 60 years and older nationwide received the influenza vaccine [44,45]. The strong influenza B wave was severely underreported in the notification data, as can be appreciated only in the comparison with wastewater influenza copy numbers.

### 3.5. Limitations of the Study

Due to technical issues and public holidays, not all time points could be analyzed. In the 2024/25 season, a higher number of observations would have been desirable, particularly in the Schmidbartlanger sewer for the comparison of spot and composite samples. The study’s sample size is relatively small, which limits statistical power and may affect the generalizability of the findings. Influenza copy numbers were often below the limit of detection at the beginning or end of the season. Dilution effects, particularly in the WWTP influent, may have decreased the viral load to levels below the limit of detection, which cannot be corrected by PMMoV normalization. The correlation with notification data only serves as a point of reference, as a massive underreporting of asymptomatic or oligosymptomatic individuals must be assumed for both influenza subtypes, and it underestimates the true burden of disease. Therefore, notification data should be interpreted alongside complementary sources, specifically wastewater data. When considering sewer specific notification data, it must be taken into account that the reported place of residence does not necessarily coincide with the location of excretion. Variables such as workplace, periods of travel, and hospitalization may exert additional influence. The challenge posed by commuter/travel patterns concerns spatial assignment, but wastewater data remains valuable because it smooths the random variability of individual case reports, providing a robust, aggregated trend for the entire catchment area. In the study, LOESS regression was applied retrospectively at the end of the influenza season rather than in real time, providing a clear overall perspective by aligning reporting data with wastewater data. For analyzes conducted during the season or forecasting, alternative methods may be required.

## 4. Conclusions

To our knowledge, this study is the first to comprehensively examine a systematic comparison of a centralized and decentralized wastewater sampling approach over three consecutive influenza seasons immediately after the COVID-19 pandemic with a total of 313 data points. The study allows the conclusion that large waves of influenza infections can be mapped well in centralized and decentralized sampling sites. Viral copy numbers correlate well with citywide case data over three seasons, while smaller outbreaks are subject to local and temporal fluctuations in the respective sampling area and are only visible on a local scale, as shown in season 2022/23. If selected properly, decentralized wastewater-based monitoring also proves to be a reliable indicator of infection trends across the entire city, a finding that could be particularly relevant for low- and middle-income countries without centralized sanitation systems. A decentralized approach enables more spontaneous and temporarily flexible wastewater sampling. Our findings suggest that a decentralized sampling approach characterized by shorter flow times within sewer systems may facilitate the detection of rapidly degrading viruses such as influenza viruses. In decentralized sampling, generally more values were measured above the LoD than in centralized sampling, which demonstrates the early detection potential of this method. Additionally, there were indications that qualified spot samples of WWTPs are also well suited for influenza A monitoring due to higher viral loads than in composite samples. The study demonstrates the suitability of composite and spot sampling methods for mapping influenza infection patterns and establishes overall feasibility for WBS. Nonetheless, the role of spot and composite sampling requires further investigation in future studies.

Only with the help of the wastewater methodology were we able to identify a reporting delay for influenza A at the beginning of the 2023/2024 season. Furthermore, only a comparison with wastewater data reveals the extent of underreporting in influenza A and B cases. As our wastewater data clearly shows, the course of the influenza season varies greatly every year. Targeted public health measures could be taken rapidly upon considering WBE data. Aware of high wastewater copy numbers, the public health service could inform the medical community to increase testing capacities at a time when an influenza season is not yet suspected, although it may already have begun. In addition, larger local outbreaks can be identified early, individualized recommendations for local district-based vaccination campaigns or masking, etc., could be given. High-resolution wastewater-based surveillance can be a complementary surveillance strategy in the event of a circulation of a pathogen with pandemic potential, such as highly pathogenic avian influenza, in order to detect first local outbreaks in selected parts of the city and apply respective measures.

## Figures and Tables

**Figure 1 microorganisms-13-02630-f001:**
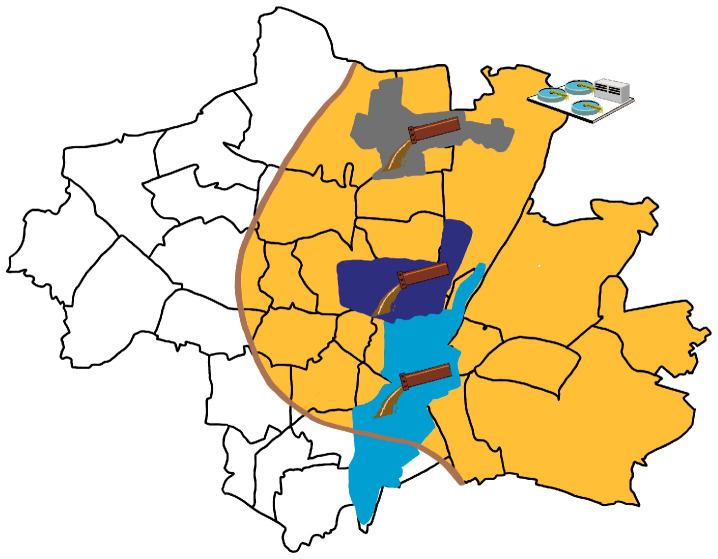
Map of sampling locations in the city of Munich, Germany. Three sewers (decentralized wastewater sampling approach) and one WWTP Gut Grosslappen (centralized wastewater sampling approach) are shown. The catchment area of the Schmidbartlanger sewer is marked in gray, Schenkendorfstr. sewer in dark blue, Gysslinger Becken sewer in light blue, and the catchment area of the WWTP Gut Grosslappen in orange. For the sewers, we were mainly provided with spot/grab samples, and for the WWTP, with 24 h composite samples.

**Figure 2 microorganisms-13-02630-f002:**
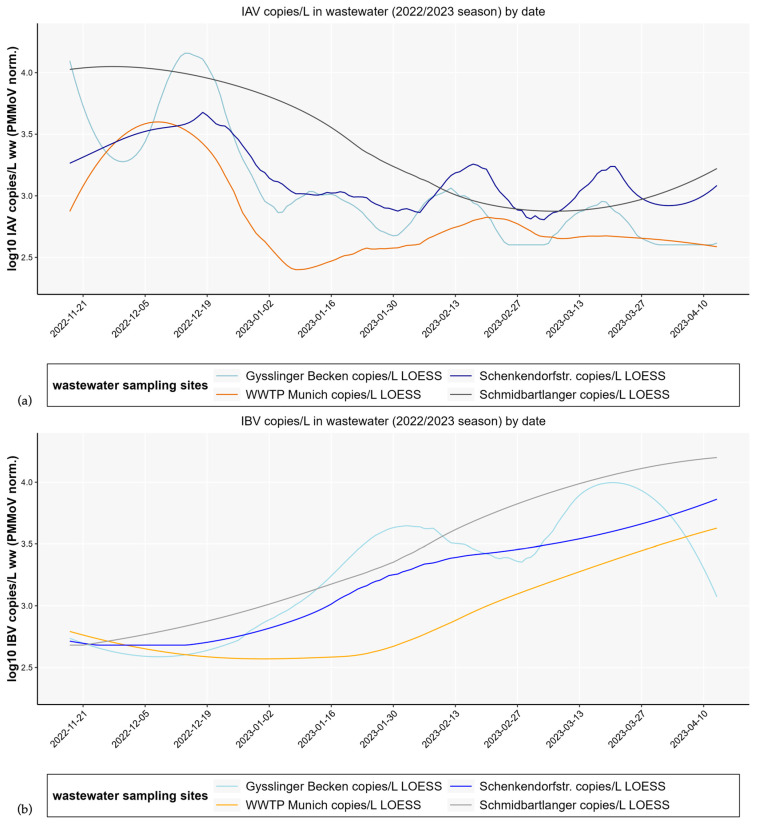
Differences in levels of PMMoV normalized IAV (**a**) and IBV (**b**) copy numbers/L smoothed by LOESS regression in wastewater depending on different sampling sites depicted for season 2022/23 with overall similar development over time for the sampling sites. Qualified spot samples of decentralized sampling sites sewer Schmidbartlanger (gray), Schenkendorfstr (dark blue), Gysslinger Becken (light blue), and a composite sample of WWTP Gut Grosslappen (orange) are shown. Sampling took place twice a week.

**Figure 3 microorganisms-13-02630-f003:**
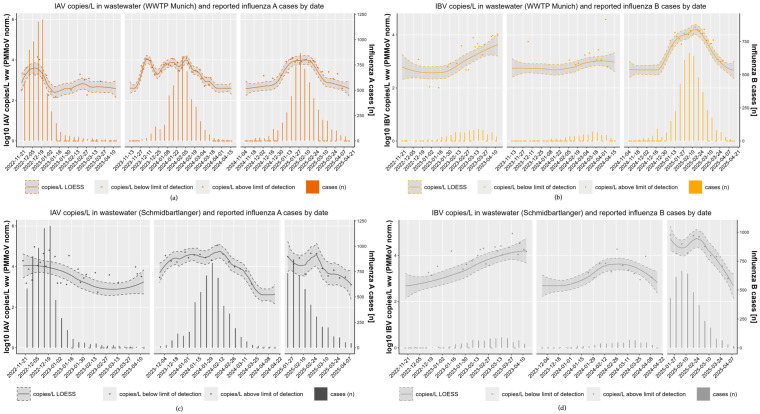
Strong variation in the course of PMMoV normalized influenza copy numbers in wastewater over time and reported influenza cases depending on the influenza season (November 2022–April 2023, November 2023–April 2024, and November 2024–April 2025). Comparison between IAV (**a**) or IBV (**b**) copy numbers in the main WWTP influent Gut Grosslappen (orange) and citywide influenza A or influenza B cases; comparison between IAV (**c**) or IBV (**d**) copy numbers in the Schmidbartlanger sewer (gray) and citywide influenza A or B cases. Samples of the main WWTP were available twice a week, samples of Schmidbartlanger sewer twice in 2022/23 and once a week in seasons 2023/24 and 2024/25. Darker dots represent influenza A viral copy numbers, lighter dots influenza B viral copy numbers in combination with corresponding LOESS smoothing lines with 95% confidence intervals (left y-axis); columns represent the numbers of known influenza A and B cases, respectively (right y-axis); viral copy numbers below the LoD are shown as hollow dots.

**Figure 4 microorganisms-13-02630-f004:**
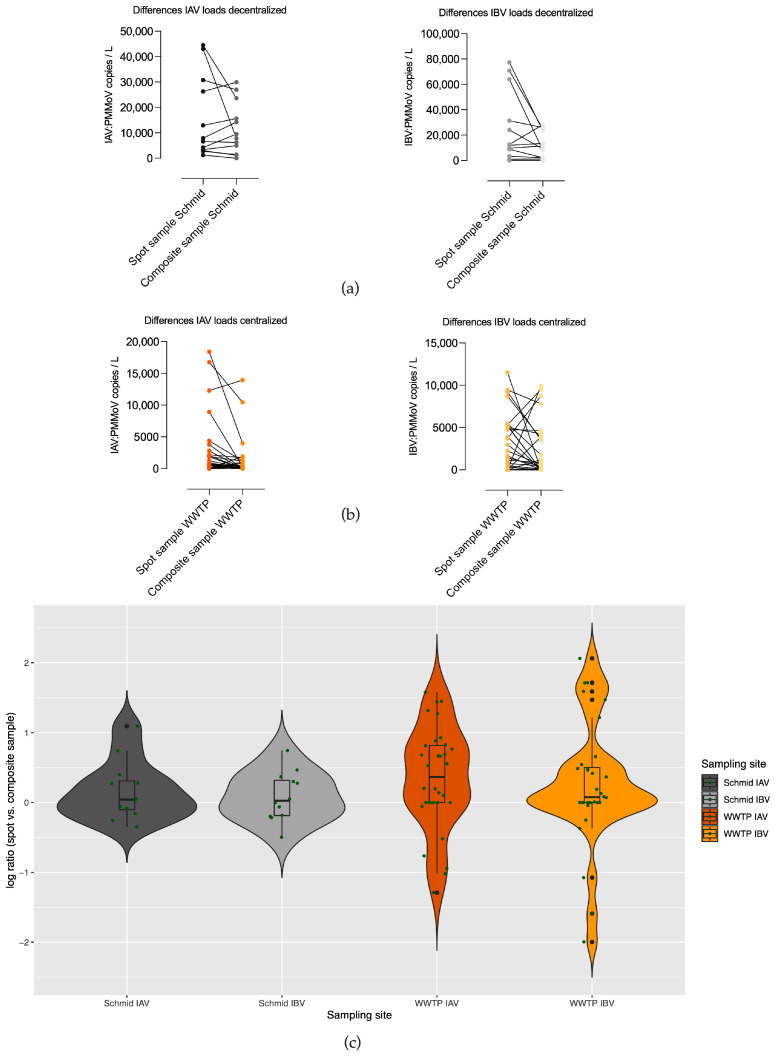
PMMoV normalized IAV and IBV viral copy numbers/L matched with respect to time, sampling type (spot or composite sampling), and location. (**a**) Differences of 12 paired, weekly viral copy numbers are plotted of a decentralized sampling strategy at Schmidbartlanger (Schmid) sewer during January–April 2025 with no evidence of higher viral copy numbers in spot or composite samples (Wilcoxon test IAV: *p* = 0.85; IBV: *p* = 0.46, *n* = 12); (**b**) differences of 32 paired, weekly viral copy numbers of a centralized sampling strategy at a WWTP during December 2022–April 2023 with indications of higher IAV copy numbers in spot samples (Wilcoxon test IAV: *p* = 0.008; IBV: *p* = 0.06, *n* = 32); concentrations below the LoD are indicated with 0 copies/L. (**c**) Violin plots including boxplots of log-ratios (spot vs. composite sample) of values as described above; green dots represent single ratios.

**Figure 5 microorganisms-13-02630-f005:**
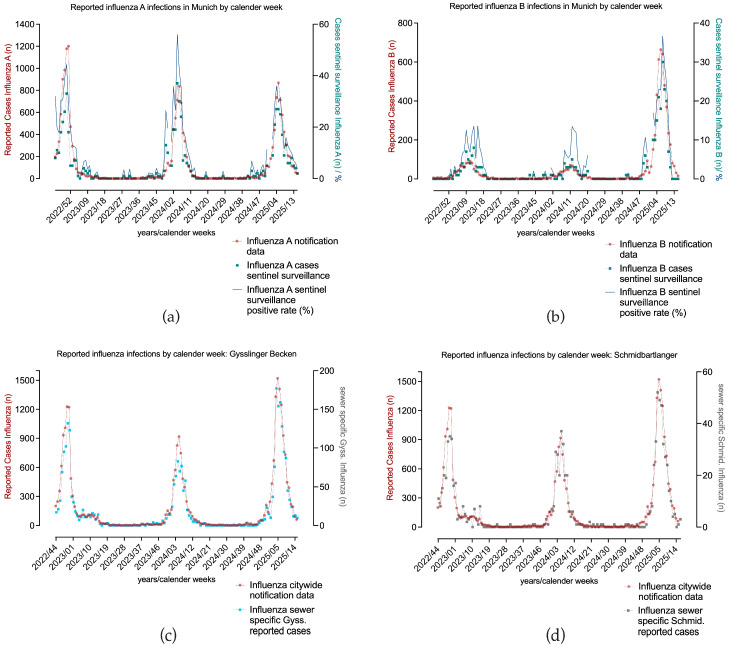
Comparison of routine influenza surveillance data based on clinical cases. Influenza A (**a**) or B (**b**) cases from citywide notification data and citywide sentinel surveillance (including positivity rate) during calendar weeks of November 2022–April 2025 are depicted; data sources correlate for influenza A (Pearson *r* = 0.90) and influenza B (Pearson *r* = 0.61); (**c**) sewer-specific influenza cases of the Gysslinger Becken sewer as a representative for a highly populated catchment area and citywide influenza cases are shown. Data sources match well (Pearson *r* = 0.99); (**d**) sewer-specific influenza cases of the Schmidbartlanger sewer and citywide influenza cases are shown. Data sources show comparable trends (Pearson *r* = 0.96).

**Table 1 microorganisms-13-02630-t001:** Selected wastewater sampling sites and their properties chosen for wastewater monitoring provided by the Munich Metropolitan Sewer Authority.

Catchment Area	Sampling Period	Connected Inhabitants	Maximum Flow [L/s]	Catchment Area Size [ha]	Max. Time Sink to Sampling [h]
Schmidbartlanger	22/23, 23/24, 24/25	67,000	180	670	5
Schenkendorfstr.	22/23	118,000	580	1050	5
Gysslinger Becken	22/23	158,000	630	1150	5
WWTP Gut Grosslappen	22/23, 23/24, 24/25	961,000	9000	20,000	12

## Data Availability

Data on influenza infections in Germany are openly available in a publicly accessible at https://survstat.rki.de/ repository (accessed on 1 September 2025). Data will be made available upon qualified request.

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
