# Peer review of "High-Resolution Wastewater-Based Surveillance of Three Influenza Seasons (2022–2025) Reveals Distinct Seasonal Patterns of Viral Activity in Munich, Germany"

_microorganisms, 2025, doi:10.3390/microorganisms13112630_

Round 1

Reviewer 1 Report

Comments and Suggestions for Authors

Dear authors,

Your manuscript is a very interesting read. It is nicely written, well structured, with straightforwards visualizations and a clear analysis. I have only a few comments:

INTRODUCTION

  • p. 3, line 100: Why would one expect a different viral decay for influenza in decentralized sampling strategies (compared to the stability mentioned above in wastewater and centralized strategies)?

METHODS

  • p. 4, lines 122-123: Why were these sampling sites chosen?
  • p. 4, lines 125-126 and Table 1: Where does this data come from?
  • p. 7, lines 244-245: How did you decide on that number (3 mm/m²) as your catchment areas differ in size, so one would expect different impacts of 3 mm/m² in each sub-sewershed.
  • p. 7, line 264: Why only Wednesdays and not Fridays or the sum of both days?
  • p. 7, line 283: "from" is missing here

RESULTS

  • p. 8, line 288, Figure 2: Why do the lines appear so different for IAV (a)? If the same smoothing technique was applied, the lines should appear equally smoothed, but this is not the case here (specifically for Schenkendorfstr.).
  • p. 9, lines 310-314: Are correlation coefficients increasing with decreasing distance to the WWTP?
  • p. 9, lines 299-300 and 319-320: With the WWTP always showing lower values, do you expect this to be due to decay, dilution, sedimentation or is this a sampling artefact as you compare grab samples of peak hours to composite samples of a whole day?
  • p. 9, lines 324-327: Good point for the early detection potential of decentralized WBS.
  • p. 10, line 352: Where does this number  (max. 5 hours) come from?

Reviewer 2 Report

Comments and Suggestions for Authors

"High–Resolution Wastewater–Based Surveillance of Three Influenza Seasons (2022–2025) Reveals Distinct Seasonal Patterns of Viral Activity in Munich, Germany" The manuscript presents a comprehensive study on wastewater-based surveillance (WBS) of influenza A and B viruses over three consecutive seasons in Munich, Germany. The study employs both centralized and decentralized wastewater sampling approaches and utilizes digital droplet PCR (ddPCR) for viral quantification. The findings highlight the potential of WBS for early detection of influenza outbreaks, tracking local transmission patterns, and identifying underreporting in traditional surveillance systems. The research is timely and relevant, given the increasing interest in wastewater epidemiology for public health monitoring. However, there are several limitations should be addressed. 
Major comments:
1. More details of the study design should be elaborated, especially for the selection of the sampling sites

2. More details regarding the determiantion of the detection limit of the ddPCR assays and the normalization method using PMMoV should be given.

3. The study suggests that spot samples may have higher viral loads than composite samples in WWTPs, but this finding is not consistent across all sampling locations. The small sample size and high variability in viral loads make it difficult to draw definitive conclusions about the superiority of one sampling method over the other. 

4. The study acknowledges several limitations, including potential biases in clinical data and the challenges associated with wastewater sampling. However, the discussion could benefit from a more detailed exploration of how these limitations might affect the study's conclusions and the broader implications for public health practice. 

5. The study's sample size, particularly for the comparison of spot and composite samples, is relatively small. This limits the statistical power and may affect the generalizability of the findings. Additionally, the inability to analyze all time points due to technical issues and public holidays results in data gaps, particularly at the beginning and end of the influenza seasons.

Minor comments:
1. Some charts, particularly Figure 2 and Figure 3, appear to have low resolution, making it difficult to discern details when enlarged. This is particularly problematic for LOESS smoothing curves and confidence intervals.

2. The labels and annotations on some charts are too small and not sufficiently detailed. For instance, Figure 2 and Figure 3 have legends and axis labels that are not immediately clear.

3. The style and format of different charts are not consistent. For example, Figure 2 and Figure 3 use different colors and line styles, which can confuse readers when comparing data across charts

Round 2

Reviewer 2 Report

Comments and Suggestions for Authors

Authors have addressed all of the reviewer's concerns